# Effect of Supplementary Light Intensity on Quality of Grafted Tomato Seedlings and Expression of Two Photosynthetic Genes and Proteins

**Hao Wei** [1,†], **Jin Zhao** [1,†], **Jiangtao Hu** [1] and **Byoung Ryong Jeong** [1,2,3,*] 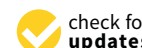

1   Department of Horticulture, Division of Applied Life Science (BK21 Plus Program), Graduate School of Gyeongsang National University, Jinju 52828, Korea; oahiew@gmail.com (H.W.); jzhao9006@gmail.com (J.Z.); jiangtaoh@yahoo.com (J.H.)
2   Institute of Agriculture & Life Science, Gyeongsang National University, Jinju 52828, Korea
3   Research Institute of Life Science, Gyeongsang National University, Jinju 52828, Korea
*   Correspondence: brjeong@gmail.com; Tel.: +82-010-6751-5489
†   These authors contributed equally to this work.

**Abstract:** Lower quality and longer production periods of grafted seedlings, especially grafted plug seedlings of fruit vegetables, may result from insufficient amounts of light, particularly in rainy seasons and winter. Supplemental artificial lighting may be a feasible solution to such problems. This study was conducted to evaluate light intensity's influence on the quality of grafted tomato seedlings, 'Super Sunload' and 'Super Dotaerang' were grafted onto the 'B-Blocking' rootstock. To improve their quality, grafted seedlings were moved to a glasshouse and grown for 10 days. The glasshouse had a combination of natural lighting from the sun and supplemental lighting from LEDs ($W_1R_2B_2$) for 16 h/day. Light intensity of natural lighting was 490 $\mu mol \cdot m^{-2} \cdot s^{-1}$ photosynthetic photon flux density (PPFD) and that of supplemental lighting was 50, 100, or 150 $\mu mol \cdot m^{-2} \cdot s^{-1}$ PPFD. The culture environment had 30/25 °C day/night temperatures, 70% ± 5% relative humidity (RH), and a natural photoperiod of 14 h as well. Compared with quality of seedlings in supplemental lighting of 50 $\mu mol \cdot m^{-2} \cdot s^{-1}$ PPFD, that of seedlings in supplement lighting of 100 or 150 $\mu mol \cdot m^{-2} \cdot s^{-1}$ PPFD improved significantly. With increasing light intensity, diameter, fresh weight, and dry weight, which were used to measure shoot growth, greatly improved. Leaf area, leaf thickness, and root biomass were also greater. However, for quality of seedlings, no significant differences were discovered between supplement lighting of 100 $\mu mol \cdot m^{-2} \cdot s^{-1}$ PPFD and supplement lighting of 150 $\mu mol \cdot m^{-2} \cdot s^{-1}$ PPFD. Expressions of *PsaA* and *PsbA* (two photosynthetic genes) as well as the corresponding proteins increased significantly in supplement lightning of 100 and 150 $\mu mol \cdot m^{-2} \cdot s^{-1}$ PPFD, especially in 100 $\mu mol \cdot m^{-2} \cdot s^{-1}$ PPFD. Overall, considering quality and expressions of two photosynthetic genes and proteins, supplemental light of 100 $\mu mol \cdot m^{-2} \cdot s^{-1}$ PPFD ($W_1R_2B_1$) would be the best choice to cultivate grafted tomato seedlings.

**Keywords:** LED; PPFD; *PsaA*; *PsbA*; Western Blot

## 1. Introduction

Light, temperature, humidity, concentration of $CO_2$, and nutrition supply are vital factors that influence photosynthesis of plants. Light intensity, quality, photoperiod, and direction influence plant growth, development, photomorphogenesis, and anatomical structure as energy sources and regulatory signals [1–3]. Previous studies have proven that light intensity can adjust formation of the chloroplast protein complex, electron transport, and quantum yield between photosystems I (PSI) and II (PSII) [4,5].

Grafting as an efficient technique can be applied to overcome the obstacles of continuous cropping and improve yield, quality, and stress resistance in various plant species including fruit trees, vegetables, and ornamental flowers [6]. In addition to rootstock, affinity of the scion, and grafted methods, the surviving rates, formation of new vascular bundles, and biological properties of grafted seedlings are also related to environmental conditions and management in healing and acclimatization processes [7]. Light-emitting diodes (LEDs) are widely used in facility horticulture as an artificial light source for production of high-quality seedlings. LEDs have shown promising foreground applications because of their advantages of being light-weight, having a small volume, having a specific wavelength, they are easy to integrate, and they have low heat dissipation. Previous studies have shown that the LEDs play a vital role in toughening grafted seedlings in healing and acclimatization processes [8,9].

Generally, stocky plug seedlings are preferred by crop growers for the convenience of transportation and to insure a higher survival ratio and enhanced development after transplanting. Each country has different standards for evaluating the quality of seedlings. In South Korea, it is generally agreed that high-quality seedlings are those without overgrowth, with short and not lodging stems and leaves, and with well-developed roots that hold the medium in the plug tray. Light intensity has been proven to affect plant height and biomass growth [10,11]. Leaf area also changes with light intensity as plants use leaves to absorb light energy [12,13]. Moreover, higher light intensities may result in more root biomass so the absorption of water and mineral elements could be promoted [14]. Therefore, seedlings with good quality could be obtained by supplementing them with optimal light intensity.

This study was carried out aimed at finding an optimal light intensity to improve the quality of two kinds of grafted tomato seedlings. After grafting, well-healed grafted seedlings were moved to and cultivated for an additional 10 days in a glasshouse, which had natural lighting and supplemental lighting. The supplementary light intensities were set at 50, 100, or 150 $\mu mol \cdot m^{-2} \cdot s^{-1}$ photosynthetic photon flux density (PPFD) supplied from mixed LEDs ($W_1R_2B_1$) to further improve the quality of seedlings. After treatment, expressions of two photosynthesis-related genes (*PsaA* and *PsbA*) and the corresponding proteins were analyzed to evaluate the capacity of photosynthesis.

## 2. Materials and Methods

### 2.1. Plant Materials

Two cultivars of tomato *(Solanum lycopersicum* L.) 'Super Sunload (SS)' and 'Super Dotaerang (SD)' were selected as the scions and tomato 'B-Blocking' as a common rootstock. Cultivars 'SS' and 'SD', widely used for grafting, were purchased from Sakata Seed Korea Co., Ltd. (Seoul, Korea) and Koregon Co., Ltd. (Anseong, Korea), respectively. The 'SS' has high fruit firmness with an average fruit weight of 220–240 g. The primary advantages of 'SS' are a low ratio of fruit deformity and resistance to high temperatures. The 'SD' is a cultivar with disease resistance and has a high sugar content, firm flesh, and a low deformity ratio. The seeds of these cultivars were sown into 40 square cell plug trays at the same time they were filled with commercial growing medium (Super Mix, NongKyung Co., Jincheon, Korea). The scion and rootstock were cleft-grafted at 20 d after sowing when both scion cultivars had two same sizes of compound leaves. After grafting, the grafted seedlings were immediately transported to a healing chamber, which had a temperature of 23 °C and relative humidity of 95% to 100% throughout the healing period. Five days later, the well-healed grafted seedlings were moved from the healing chamber to the greenhouse for light treatment when the average plant height was about 8 ± 0.7 cm and each seedling had two and a half true leaves.

### 2.2. Light Treatments

Mixed LEDs ($W_1R_2B_1$) (Custom made, Sung Kwang LED Co., Ltd., Incheon, Korea), which were selected as the best artificial light source in precedent research [15], were used as the sole light source. The grafted seedlings were cultivated for an additional 10 d in a glasshouse with a daily light integral (DLI) of 11.59 $mol \cdot m^{-2} \cdot d^{-1}$ from the sun and supplemental lighting for 16 h/d. The supplementary

light intensities were set at 50, 100, or 150 μmol·m$^{-2}$·s$^{-1}$ PPFD (henceforth shortened as *50*, *100*, and *150*). The DLIs of the three supplementary light treatments were 2.88, 5.76, and 8.64 mol·m$^{-2}$·d$^{-1}$, respectively. The distance between the LEDs and seedlings was 50 cm. Light intensity was measured at the horizontal position at the seedling stem. The culture environment had 30/25 °C day/night temperatures, 70% ± 5% relative humidity (RH), with a natural photoperiod of 14 h. A portable photo/radiometer (Type specification: HD-2102.2, Delta, OHM, Padova, Italy) was used to measure the photon flux density. Different light intensities were controlled by adjustable electric currents.

### 2.3. Stomatal Conductance

Mature and expanded leaves were selected for the measurement of stomatal conductance. Measurements were carried out during the day (9:00–10:00) on a porometer (Type specification: Sc-1 Porometer, Decagon Devices Inc., Pullman, WA, USA).

### 2.4. Quantitative Real-Time PCR Analysis

Tomato leaves were ground under RNase-free conditions after being frozen in liquid nitrogen. An Easy-Spin total RNA extraction kit (Type Specification: iNtRON Biotechnology, Seoul, Republic of Korea) and a GoScript Reverse Transcription System (Specification: Promega, Madison, WI, USA) were used for RNA extraction and cDNA synthesis, respectively, following the manufacturer's instructions. The Rotor-Gene Q detection system (Qiagen, Hilden, Germany) was applied in the examination of gene expression. Reaction volumes (20 μL) contained 1 μL cDNA, 0.5 μL primer (10 μM), 10 μL of 2× AMPIGENE qPCR Green Mix Lo-ROX (Enzo life Sciences Inc., Farmingdale, NY, USA), and 8 μL ddH$_2$O. The 2$^{-\triangle\triangle Ct}$ method was applied to determine the relative gene expressions, and the *18S* gene was used as housekeeping gene. All primers used are shown in Supplementary Table S1.

### 2.5. Protein Extraction and Western Blotting

Protein extraction was conducted following the method described elsewhere [16], with slight modifications. An extraction buffer was used to homogenize approximately 500 mg tomato leaves. The buffer contained 100 mM Tris-HCl (pH 8.0), 0.5 mM EDTA·Na$_2$, 1% (v/v) polyvinylpyrrolidone, 10 mM β-mercaptoethanol, and 200 mM sucrose. At 4 °C, the mixture was centrifuged at 12,000 rpm for 10 min. Subsequent to determination of protein concentrations, the supernatant (10 μL) was mixed together with protein loading buffer and then separated on 12% SDS-PAGE gel. After that, proteins were moved to a polyvinylidene fluoride (PVDF) blotting membrane (Type specification: Millipore, Bedford, MA, USA) using an electrotransfer instrument (Type specification: Bio-Rad, Hercules, CA, USA). Following electrotransfer, the PVDF blotting membrane was transferred into 5% nonfat skimmed milk, dissolved in tris buffered saline tween (TBST) overnight at 4 °C to prevent nonspecific adsorption, washed with TBST, and incubated by using the following polyclonal primary antibodies: 1:3000 dilution of anti-PsaA (Specification: Agrisera AS06 172, Vännäs, Sweden) for PsaA, 1:10,000 dilution of anti-PsbA (Specification: Agrisera AS05 084, Vännäs, Sweden) for PsbA, and 1:10,000 dilution of anti-RbcL (Specification: Agrisera AS03 037, Vännäs, Sweden) for loading control overnight at 4 °C, respectively. After that, PVDF blotting membranes were incubated in 1:10,000 dilution of HRP (horseradish peroxidase)-linked anti-rabbit 1gG (Bethyl A120 101P, Montgomery, TX, USA) for 1 h at ambient temperature as a secondary antibody before visualization of immunoreactive proteins using Clarity™ Western ECL Substrate (#6883, Bio-Rad, Hercules, CA, USA) on an iBright™ Imaging Systems (Type specification: Thermo Fisher Scientific, Waltham, MA, USA).

### 2.6. Data Collection and Analysis

After supplemental lighting treatments for 10 d, this study measured the following parameters: lengths of scion and root, diameters of scions, number of leaves, leaf length, leaf width, leaf area and thickness, chlorophyll (SPAD), fresh weight of root, dry weight of root, specific weight of leaf, and dry weight to height ratio of scion.

The experiment was carried out in three replicates with a design of randomized complete block, using 9 seedlings in each replication. Locations of replications were randomly mixed to eliminate position effect within a controlled environment. One-way analysis of variance ANOVA by SAS program Statistical Analysis System (V.9.1, Cary, NC, USA), was applied to statistically analyze the collected data. Duncan's multiple range test was adopted to test significant differences between different means.

## 3. Results

### 3.1. Growth and Development of Tomato Seedlings as Affected by Supplemental Lighting Treatment in the Glasshouse

After cultivation in a glasshouse for 10 d, growth and development parameters of the grafted tomato seedlings were measured. Results of the measurements showed that the characteristics of the seedlings' morphologies were significantly influenced by light intensities. The two cultivars of grafted tomato cultivated in *100* and those in *150* showed the greatest scion length and root fresh weight. No significant differences were discovered in scion stem diameter and leaf length between supplementary lighting of *50*, *100*, or *150*. Leaf area and width, number of leaves, and root length of 'SS' were significantly greater in supplementary lighting of *100* and *150* than in supplementary lighting *50*. The greatest root length was observed in light intensity of *150* in 'SD'. In general, there were little differences between supplementary lighting of *100* and *150*. Compared with supplementary lighting of *50*, both of these two treatments significantly improved quality of the two types of grafted tomato seedlings (Table 1).

Dry weight of shoot and dry weight of root are shown in Figure 1. Considering these two weights, there were no significant differences between light intensity of *100* and light intensity of *150*. However, compared to the treatment of light intensity of *50*, these two treatments significantly improved dry weight of the two types of tomato cultivars.

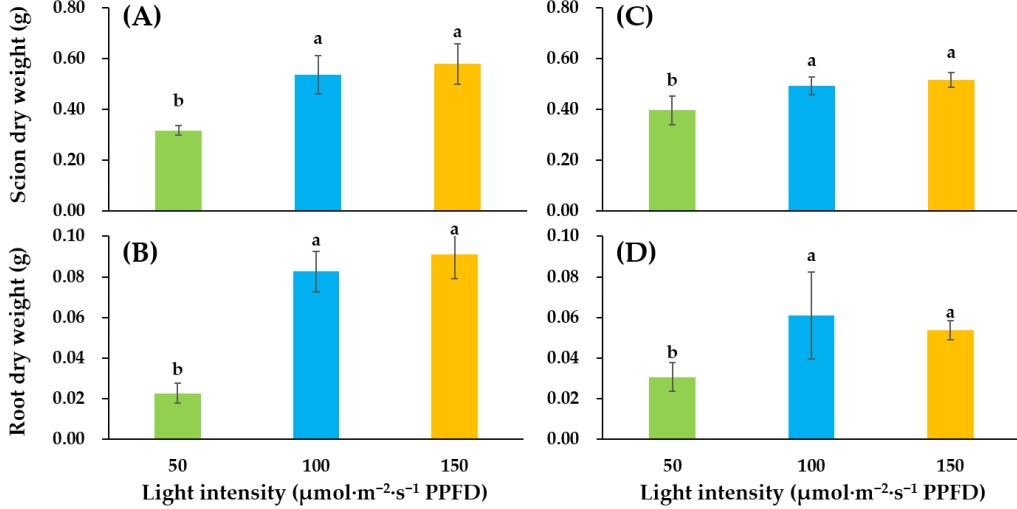

**Figure 1.** Influence of supplementary lighting intensity on dry weight of shoots and dry weight of roots of two cultivars of tomato 'Super Sunload' ('SS') (**A**,**B**) and 'Super Dotaerang' ('SD') (**C**,**D**) grafted onto the 'B-Blocking' rootstock. Error bars represent SEs of nine biological replicates (*n* = 9). According to the Duncan test, lowercase letters at $p \leq 0.05$ indicate significant differences between different treatments.

**Table 1.** Influence of supplemental light intensity on growth and development of grafted tomato seedlings grown for 10 d.

| Cultivar (C) | Light Intensity (LI) ($\mu$mol·m$^{-2}$·s$^{-1}$ PPFD) | Scion | | | | | | | | Root | |
| | | Length (cm) | Stem Diameter(mm) | No. | Length(cm) | Width (cm) | Area (cm$^2$) | Thickness (mm) | Chlorophyll (SPAD) | Length (cm) | Fresh Weight (g) |
|---|---|---|---|---|---|---|---|---|---|---|---|
| 'SS' | 50 | 13.2 b $^z$ | 5.13 a | 7.8 b | 14.3 | 8.7 b | 61.9 b | 0.35 b | 44.1 b | 8.6 b | 0.4 b |
| | 100 | 19.1 a | 5.31 a | 8.8 a | 15.0 | 9.0 ab | 73.6 ab | 0.39 b | 47.6 ab | 17.2 a | 1.1 ab |
| | 150 | 20.2 a | 5.69 a | 9.3 a | 15.8 | 9.9 a | 84.5 a | 0.47 a | 50.3 a | 18.6 a | 1.9 a |
| 'SD' | 50 | 16.3 b | 5.52 a | 8.7 a | 15.3 | 8.8 a | 70.0 a | 0.36 a | 39.5 c | 9.9 c | 0.4 b |
| | 100 | 20.1 a | 5.63 a | 9.2 a | 15.7 | 9.3 a | 69.9 a | 0.39 a | 48.2 a | 14.6 b | 1.0 a |
| | 150 | 19.5 a | 5.96 a | 9.8 a | 16.0 | 8.7 a | 68.1 a | 0.39 a | 42.6 ab | 16.9 a | 1.0 a |
| F-test | | | | | | | | | | | |
| LI | | *** $^y$ | * | *** | NS | * | * | ** | * | *** | *** |
| C | | NS | * | * | NS | NS | NS | NS | NS | NS | NS |
| LI * C | | NS | NS | NS | NS | NS | ** | NS | NS | NS | NS |

$^z$ Means followed by same letter (s) within a column are not significantly different ($p \leq 0.05$). $^y$ NS, not significant. *, **, and ***, significant at $p$ = 0.05, 0.01, or 0.001, respectively.

Growing compact and sturdy seedlings is an objective of high-quality seedling production. Dry weight to height ratio of scion (WHR) was conducted to indicate the quality of seedlings. For 'SS', supplementary lighting of *100* and *150* significantly improved WHR as compared to supplementary lighting of *50*. There were no significant differences in WHRs between supplementary lighting of *100* and *150*. For 'SD', the greatest WHRs showed in supplementary light of *150*. For both tomato cultivars, the specific leaf weight (a leaf fresh weight per unit leaf area) was significantly improved in light intensities of *100* and *150* as compared to light intensity of *50*. There were no significant differences in the specific leaf weight between light intensity of *100* and 150 (Figure 2).

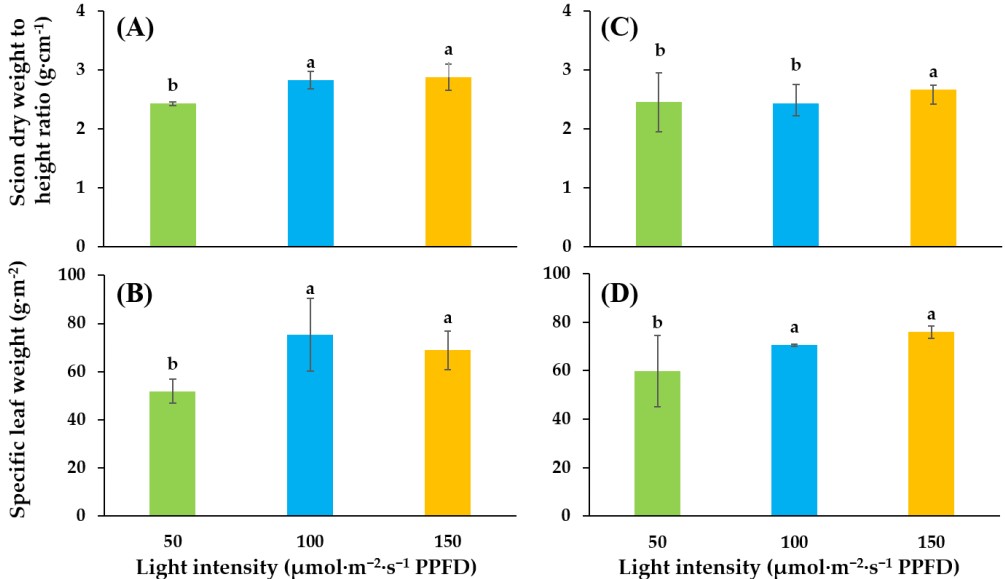

**Figure 2.** Influence of supplementary lighting intensity on dry weight to height ratio of scion and specific leaf weight of two cultivars of tomato 'SS' (**A**,**B**) and 'SD' (**C**,**D**) grafted onto 'B-Blocking' rootstock. The error bars represent SEs of biological replicates (*n* = 9). According to the Duncan test, lowercase letters at *p* ≤ 0.05 indicate significant differences between different treatments.

Moreover, we determined the effect of lighting intensity on stomatal conductance of the two tomato cultivars. Results showed, compared with light intensity of *50,* light intensity of *100* increased stomatal conductance of the two cultivars. Interestingly, stomatal conductance in light intensity of *150* was similar to that in light intensity of *50* treatment (Figure 3).

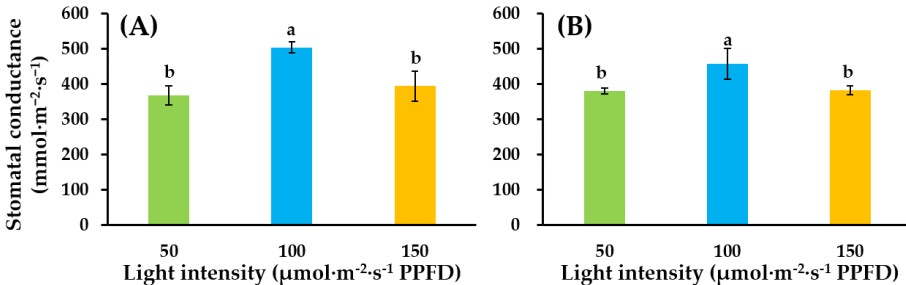

**Figure 3.** Influence of supplementary lighting intensity on stomatal conductance of two cultivars of tomato 'SS' (**A**) and 'SD' (**B**) grafted onto the 'B-Blocking' rootstock. The error bars represent SEs of three biological replicates (*n* = 3). According to the Duncan test, lowercase letters at *p* ≤ 0.05 indicate significant differences between treatments.

Figure 4 illustrated seedling morphologies of 'SS' and 'SD' after treating for 10 d by using supplementary lighting with a different light intensity. It showed that tomato seedlings grown in supplementary lighting of *100* and *150* had more biomass and were sturdier.

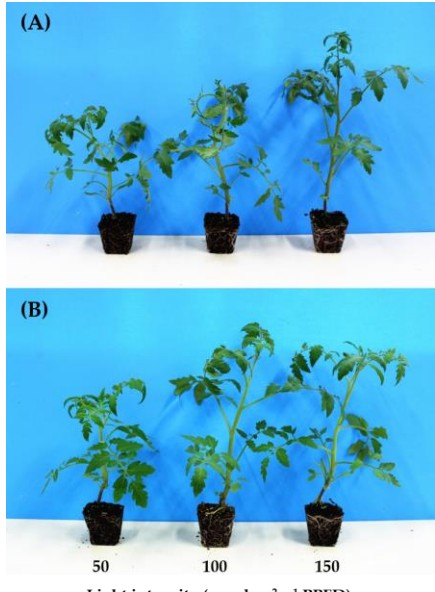

**Figure 4.** Influence of supplementary light intensity on morphology of 'SS' (**A**) and 'SD' (**B**) after 10 d of cultivation.

*3.2. Expressions of Photosynthesis-Related Genes*

Light intensity significantly influenced *PsaA* and *PsbA* genes (Figure 5). Light intensity of *100* enhanced expressions of *PsaA* and *PsbA*. Relative expressions of *PsaA* in 'SS' and 'SD' were 1.37- and 2.27-fold, while that of *PsbA* was 4.01- and 10.29-fold, as compared to that in the light intensity of *50*. However, a dramatic decrease in expression was observed in light intensity of *150* in 'SS' and 'SD'. Particularly, the relative expression of *PsaA* in 'SS' was only 0.44-fold compared to the expression level in *50*.

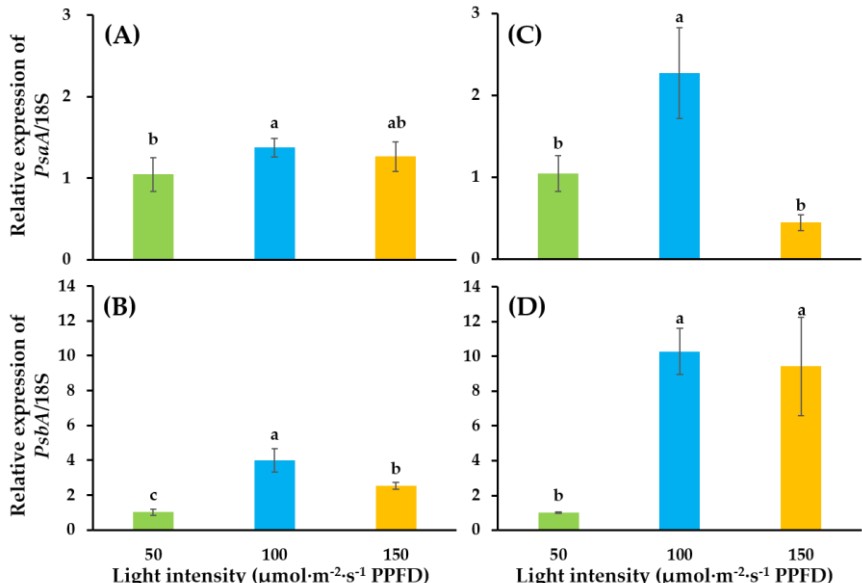

**Figure 5.** Relative expressions of *PsaA* and *PsbA* genes in 'SS' (**A,B**) and 'SD' (**C,D**) grafted onto the 'B-Blocking' rootstock. The bars among treatments represent the mean ± SEs of technological replicates (*n* = 3). According to the Duncan test, lowercase letters at $p \leq 0.05$ indicate significant differences between different treatments.

### 3.3. PsaA and PsbA Immunoblots

The *PsaA* gene is located in chloroplast genome (cpDNA). It encodes a core protein of a protein-pigment complex in PSI [17] and encodes D1 protein, which is a core component of PSII [18]. Figure 6 shows the protein expressions of PsaA and PsbA in 'SD' and 'SS' cultivated in supplementary lighting with different light intensities (*50*, *100*, or *150*). For both tomato cultivars, expression levels of PsaA and PsbA in light intensities of *100* and *150* increased as compared to those in light intensity of *50*. The highest expression levels of the PsaA and PsbA proteins appeared in light intensity of *100*.

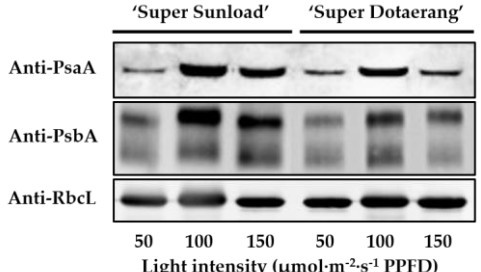

**Figure 6.** Expressions of PsaA and PsbA protein in 'SD' and 'SS' grafted onto the 'B-Blocking' rootstock. Equal loading was verified by the Rubisco large subunit (RbcL) level.

## 4. Discussion

Among regulatory factors for light condition, light intensity as an essential element is involved in regulation of growth, yield, and nutritional quality of vegetables [19]. Previous studies have proven that increased light intensity within a certain range may promote growth in lettuce [20,21]. Light condition is the foundation and guarantees healthy growth and yield of tomatoes [22]. Recent studies have shown that tomato seedlings under low-light conditions showed elongated internodes and petioles as well as a larger leaf area, which were caused by a shade-avoidance response [23,24]. Low-light conditions also affect the differentiation time and quality of flower buds in tomatoes [25]. For tomato cultivars 'Matador' and 'Blizzard', plant height, leaf width, number of leaves, and dry weight of aerial plant parts significantly improved due to increased levels of supplementary light (30, 60, and 90 $\mu$mol·m$^{-2}$·s$^{-1}$ PPFD) [26]. However, these findings do not mean the higher light intensities are always better. Tomato seedlings treated with 150 $\mu$mol·m$^{-2}$·s$^{-1}$ PPFD using white and blue LEDs showed continuous light-induced injuries, and the degree of the injury was more serious under white and blue LEDs than under orange and red LEDs of the same intensity [27]. This study showed that, compared with stem diameter of grafted tomato seedlings in supplementary lighting of *50*, that of grafted tomato seedlings increased in supplementary lighting of *100* and *150*. Leaf width and leaf thickness were also greater with treatments of light intensity of *100* and *150* (Table 1). Higher over-ground biomass and activeness of antioxidant enzymes were discovered in treatments with increased light intensity (100, 200, and 400 $\mu$mol·m$^{-2}$·s$^{-1}$ PPFD) in lettuce [28]. For *Nicotiana tabacum*, seedling biomass, soluble sugar concentrations, and primary root growth increased dramatically with light intensity of 300 $\mu$mol·m$^{-2}$·s$^{-1}$ PPFD as compared to 60 $\mu$mol·m$^{-2}$·s$^{-1}$ PPFD [14]. It had been proven that low light intensity decreased above- and below- ground biomass in radish [29]. The results also showed, compared with light intensity of *100* and *150*, dry weight of scion and dry weight of root were much lower in light intensity of *50* (Figure 1). Additionally, light intensity influences the development of stomata as well. According to the study of Lee [30], the stomatal conductance and the number of stomata per unit area of leaf increased with light intensity, increasing up to 60 $\mu$mol·m$^{-2}$·s$^{-1}$ PPFD in *Withania somnifera*. However, width and length of stomata declined in light intensity of 90 $\mu$mol·m$^{-2}$·s$^{-1}$ PPFD. These findings were consistent with our results in Figure 3. The *100* promoted stomatal conductance of both tomato cultivars, while *150* affected it adversely.

The aim of supplementary lighting in greenhouses are as follows: to increase the light intensity in greenhouse and enable crops to receive proper light, to provide crops with light with specific

wavelengths, to alter the time of flowering and fruiting, and to increase time for photosynthesis via changing the lighted period [31,32]. Advantages of light-emitting diodes (LEDs) include their small volume, longer service life, energy savings, and use of a single wavelength, which are efficient to encourage plant growth by producing the type of light with the specific wavelength needed by plants [33]. Studies have shown that red and blue lights absorbed by plants account for 90% of total light absorption [34], which are involved in regulating gene expression in photomorphogenesis [35]. Our previous study showed that compared with high pressure sodium (HPS), far-red (FR), metal halide (MH), and white LEDs (W), mixed LEDs ($W_1R_2B_1$) as a supplementary light source had the best effect to improve the quality of grafted tomato seedlings [15]. The application of monochromatic light may cause physiological defects of plants such as disturbance of photosynthetic mechanisms [36], damage to the granule and thylakoid [37], and restrained synthesis of chlorophyll [38], which can be overcome or relieved by the mixture of red and blue light.

Photosynthesis is one of the supreme chemical reactions in plants. Photoreaction takes place in the chloroplast thylakoid membrane. Two light energy-driven systems, PSI and PSII, synergistically function in primary energy conversion reactions. Driven by light, plants and other photoautotrophs are able to synthesize carbon compounds directly from $CO_2$ and $H_2O$ and release $O_2$. Processes of consumption and formation in photosynthesis are related to photosynthetic phosphorylation coupled with electron transport. Electrons generated from splitting of $H_2O$ by the oxygen-evolving complex (OEC) are used to reduce $NADP^+$ to NADPH. At the same time, synthesis of ATP is driven by a transmembrane electrochemical proton gradient [39,40]. The PSII, a multi-subunit protein-pigment complex, is associated with the photolysis of water and synthesis of ATP [41]. Within the core of the complex, PsbC and PsbB are bond to the photosynthetic pigments and transfer the excitation energy to reaction center proteins D1 (Qb, PsbA) and D2 (Qa, PsbD) [42]. D1 dimerize with D2 as a heterodimer. Studies have suggested that synthesis of D1 is coordinated with linear photosynthetic electron transport as well as the relative activities of PSI and PSII [43]. Light has an important role in the translation of Dl protein [44]. The PSI complex contains reaction center P700, as well as five electron transfer centers (A0, A1, FX, FA, and FB) bound to PsaA and PsaB proteins [45]; it mediates electron transfer and is involved in light-driven conversion of $NADP^+$ to NADPH [46]. The two largest polypeptides, PsaA and PsaB, carrying P700 are believed to form the center Fx through dimerization [47]. In higher plants, the genes *PsaA* and *PsaB* encode the two polypeptides respectively and are located adjacent to each other in the plastid genome [48].

A previous study proved that expression of the *PsaA* gene product during rice plastid development was light-regulated on the translational or post-translational level [49]. In *Synechococcus*, blue light photoreceptor regulates expressions of *PsbA* genes [50]. Moreover, in *Synechocystis* the transcription levels of promoters of *Psba* gene changed in response to dark and light conditions [51].

Expression levels of *PsaA* and *PsbA* genes in light intensities of *100* and *150* increased as compared to those in light intensity of *50* in both cultivars. Compared with those in light intensity of *100*, lower expression levels of genes appeared in light intensity of *150* (Figure 5). Western blotting showed the same tendency of variation in protein expression as shown in Figure 6. Previous studies pointed that high light intensities caused degradation and phosphorylation of PSII core protein and damaged the photosynthetic machinery [52]. It could be speculated that the photosynthetic apparatuses suffered slight damage in light intensity of *150*.

Although light is the basic energy for photosynthesis, light that is too strong will cause light stress for plants, which has side effects for photosynthesis, especially under low temperatures, drought, or other unfavorable conditions [53]. In 1956, Kok firstly proposed the concept of photoinhibition and defined it as 'the photochemical inactivation of complete pigment complexes or photosynthetic units' [54]. Powles (1984) pointed that excessive light energy absorbed by the photosynthetic apparatus caused photoinhibition with the characteristic of reduced photosynthetic capacity [55]. PSII is very sensitive to strong light, and it is the main occurrence site of photoinhibition [40]. There are two mechanisms of photoinhibition in PSII, acceptor-side photoinhibition and donor-side

photoinhibition [56]. In the former situation, the hindered assimilation of $CO_2$ will cause the reduction of the plastoquinone pool. Consequently, the double-reduced form of $QA^-$ accumulates and promotes the formation of the triplet state of the primary donor ($^3$P680). The $^1O_2$ generated from the reaction between $^3$P680 and $O_2$ will cause damage to proteins and pigments, especially the amino acid residue sites in the D1 protein. In another mechanism, electron transport is impaired due to the hindered oxidation of water and prolonged half-life period of P680$^+$ resulting in high oxidation potential on the donor side. Adjacent proteins and pigments are damaged as a consequence [57].

In 1994, Terashima firstly reported photoinhibition in PSI [58]. High-light intensity [59,60], low temperatures [61], and fluctuating light conditions [62,63] will cause PSI photoinhibition. Studies have shown that changes of reducing state and production of ROS (reactive oxygen species) and hydroxyl peroxide on the acceptor side are the main causes of PSI photoinhibition [64,65].

Plants have developed various mechanisms to prevent photoinhibition under high-light intensities because of their long-term evolution and development. The main mechanisms of photoprotection are as follows: to reduce the light absorption via movements of leaf and chloroplast [66]; to increase photosynthetic efficiency via increased levels of the photosynthetic electron transport carriers and contents of the photosynthetic key enzymes [67,68]; to protect the photosynthetic apparatus by photochemical and non-photochemical pathway such as the xanthophyll cycle, photorespiration, and the Mehler reaction [69–71]; to strengthen the active oxygen eliminating enzyme system [72,73]; and to distribute excitation energy and heat dissipation equally via state transition between PSII and PSI [74,75].

Usually, sun and shade plants possess different morphological, physiological, and biochemical characteristics to adapt to different light conditions. Chloroplasts with distinctive characteristics will contribute to levels of photosynthesis. As compared to sun plants, shade plants have higher thylakoid stack areas, fewer grana stacks, and smaller widths of grana stacks to inhibit degradation of the D1 protein [76]. The levels of photosynthetic electron transport carriers and activities of photosynthesis-related enzymes, especially Rubisco, are higher in sun plants [77,78]. When light intensity is changed, the activities of RuBP and Rubisco do respond to the fluctuating environment [79]. Studies in mulberry showed that the leaves, which were at the daily maximum photosynthetic state, showed higher reversibility of photosynthetic quantum conversion, confronting the environment with changing light intensity [80]. After being transferred to an intercropping shaded environment, the lodging-resistant soybean genotypes showed elongated main stems and internodes as well as decreased biomass to the length of the stem ratio. The reasons may be due to downregulated gene expressions related to biosynthesis of structural carbohydrates [81]. Additionally, shade conditions influenced the photosynthetic rate by altering the chlorophyll fluorescence performance in soybean [82] and *Athyrium pachyphlebium* [83]. Shade treatment changed concentrations of chlorophyll and chlorophyll a/b ratios in soybeans [84] and wheat [85]. These studies suggested that light capture and utilization, PSII activity, and the transfer of electron and energy between PSII and PSI may involve the regulation of photosynthetic capacity in shade and light conditions. Phenotypic plasticity of plants refers to the ability to conform their morphology and physiology to the environment [86]. Even though studies have shown that the phenotypic plasticity of some shade plant species is lower [87,88], the mechanism of optimizing light energy capture should be lucubrated. To illustrate that, the mechanism of photosynthesis in natural growth condition with fluctuating light can help plants to adapt to sun and shade conditions and increase yields as a consequence [86]. Additionally, more attention should be paid to the relationship between genotype plasticity and productivity as well as the intraspecific variation in plasticity to fluctuating light [89].

Light required by plants is related to different species, cultivars, growth stages, developmental stages, and environmental factors as well as target of production [19]. Although light has a vital function in photosynthesis, excess light may depress photosynthetic efficiency, which is referred to as photoinhibition [55]. Therefore, to maximize economic benefits of grafted seedlings with high quality and quantity, detailed studies on optimizing light intensity are in urgent need.

## 5. Conclusions

When moved to and cultivated in a glasshouse having supplementary lighting, light intensities of *100* and *150* had significantly improved qualities of grafted tomato seedlings as compared to light intensity of *50*. However, considering improvement of quality of grafted seedlings, there were no significant differences between light intensities of *100* and *150*. Owing to enhanced photosynthesis, the upregulated gene expression levels of *PsaA* and *PsbA* and the corresponding protein expression levels indicated that light intensity of *100* might be the most suitable supplementary lighting to improve the quality of grafted tomato seedlings. That is, supplementary lighting with a light intensity of *100* would be the best choice to improve the quality of grafted vegetable seedlings when both power consumption and economic benefits are taken into account. This study provides new thoughts on supplemental lighting strategies that might be applied in greenhouses and lays the foundation for further studies on the utilization of light energy during early development stages of tomato seedlings.

**Supplementary Materials:** The following are available online at http://www.mdpi.com/2073-4395/9/6/339/s1, Table S1: Sequences of Primers used in the expressions of photosynthesis-related genes.

**Author Contributions:** Conceptualization, B.R.J.; Methodology, B.R.J. and H.W.; Formal Analysis, H.W., J.Z., and J.H.; Resources, B.R.J.; Data Curation, H.W. and J.Z.; Writing—Original Draft Preparation, H.W.; Writing—Review & Editing, B.R.J.; Project Administration, B.R.J.; Funding Acquisition, B.R.J.

**Funding:** This research was funded by Korea Institute of Planning and Evaluation for Technology in Food, Agriculture, and Forestry, Project No.319008-01. H.W., J.Z., and J.H. were supported by a scholarship from the BK21 Plus Program, Ministry of Education, Republic of Korea.

**Acknowledgments:** This study was carried out with support from the Korea Institute of Planning and Evaluation for Technology in Food, Agriculture, and Forestry (Project No.319008-01). H.W., J.Z., and J.H. were supported by a scholarship from the BK21 Plus Program, Ministry of Education, Republic of Korea.

**Conflicts of Interest:** The authors declare no conflict of interest.

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
