# Peer review of "Effect of Supplementary Light Intensity on Quality of Grafted Tomato Seedlings and Expression of Two Photosynthetic Genes and Proteins"

_agronomy, doi:10.3390/agronomy9060339_

Reviewer 1 Report

Teoretical and practical reasons for the experiments are very reasonable. However the paper should be better introduced and the aims are clearly stated. Authors could write about reasons and benefits of suplemented light intensity. On the other hand authors could pay attention to photoinhibition and processe of fotoprotection of photosynthesis under high light treatment.

The methods are sufficiently described. The agronomic characteristics of the genotypes were not well described in the manuscript.

Authors could discuss about potential mitigation mechanisms from light stress. What are future suggestion on the level of breeding of the genotypes with higher plasticity? Authors could underline new original findings.

Overall, the study is of good quality and the results are inovative.

Innovation of the paper is convected to the journal. Conclussions and pesrpectives of the research could be reconsidered.

Authors could discuss more about complexity of the regulation of light conversion and performance of photosynthetic apparatus under high light and shade conditions.

I invite authors to add additional references to support the text (in introduction and discussion sections):

Huang, W., Zhang, S.B., Liu, T.: Moderate photoinhibition of photosystem II significantly affects linear electron flow in the shade-demanding PlantPanax notoginseng. Front. Plant Sci. 9, 2018, 250–256

Valladares, F., Niinemets, U.: Shade tolerance, a key plant feature of complex nature and consequences. Annu. Rev. Ecol. Evol. Syst. 39, 2008, 237–257

Husajn S., Guopeng C., Shafiq I., Asghar M.A., et al.: Shade effect on carbohydrates dynamics and stem strength of soybean genotypes. Environmental Experimental Botany, 162, 2019, 374-382 https://doi.org/10.1016/j.envexpbot.2019.03.011              

Author Response

(All the changes were highlighted in green in the manuscript)

1. Teoretical and practical reasons for the experiments are very reasonable. However, the paper should be better introduced and the aims are clearly stated. Authors could write about reasons and benefits of suplemented light intensity. On the other hand, authors could pay attention to photoinhibition and processe of fotoprotection of photosynthesis under high light treatment.

Response: Reasons and benefits of supplemented light intensity have been added line 262-265. The brief introduction of photoinhibition has been added in line 305-323. And the processes of photoprotection of photosynthesis under high light treatment are summed up in line 324-332.

2. The methods are sufficiently described. The agronomic characteristics of the genotypes were not well described in the manuscript.

Response: The agronomic characteristics of the two tomato cultivars has been added in line 76-80.

3. Authors could discuss about potential mitigation mechanisms from light stress. What are future suggestion on the level of breeding of the genotypes with higher plasticity? Authors could underline new original findings.

Response: Related content has been added in line 324-332 & 346-353.

4. Overall, the study is of good quality and the results are inovative.

Innovation of the paper is convected to the journal. Conclussions and pesrpectives of the research could be reconsidered. Authors could discuss more about complexity of the regulation of light conversion and performance of photosynthetic apparatus.

Response: Related content has been added in line 333-346.

5. I invite authors to add additional references to support the text (in introduction and discussion sections):

Huang, W., Zhang, S.B., Liu, T.: Moderate photoinhibition of photosystem II significantly affects linear electron flow in the shade-demanding PlantPanax notoginseng. Front. Plant Sci. 9, 2018, 250–256

Valladares, F., Niinemets, U.: Shade tolerance, a key plant feature of complex nature and consequences. Annu. Rev. Ecol. Evol. Syst. 39, 2008, 237–257

Husajn S., Guopeng C., Shafiq I., Asghar M.A., et al.: Shade effect on carbohydrates dynamics and stem strength of soybean genotypes. Environmental Experimental Botany, 162, 2019, 374-382 https://doi.org/10.1016/j.envexpbot.2019.03.011.

Response: Three articles have been cited in discussion in line 283, 346,347, & 351.

Reviewer 2 Report

 Authors investigated in this article the influence of the level of light intensity of supplemental LEDs lighting on quality of grafted tomato seedlings. They evaluated shoot growth and expression of photosynthetic genes and proteins appeared in grafted tomato seedlings after three levels of supplemental lighting as a treatment for 10 days. Authors concluded that the quality of grafted tomato seedlings was enhanced by the treatment with the optimal PPFD. The findings look helpful for grower of tomato, however, more information should be described in materials and methods to clarify their availability and reproducibility of the result. The meaning of a word, quality, of grafted tomato seedlings was obscure. What is required for grafted tomato seedling to develop well or mature? I could not understand if the assessment of the quality were provided sufficiently with the data appeared in the manuscript. Therefore, the manuscript would be improved if authors could address concerns raised below.

In plant materials, how do I confirm that two cultivars of scions purchased and used for grafting to connect with rootstocks had the same stage at plant development?

In light treatments, the condition of healing and the initial state of grafted seedling before cultivation in a greenhouse with light treatments should be stated.

In light intensity. How much total PPFD did grafted seedlings gain from natural and supplemental lighting? How far from seedlings were LEDs set in a greenhouse, a distance between seedlings and LEDs, and where were PPFD measured at?

Line 80 and 82, spaces should be inserted between the number and unit: 16 h/day, 25/35 Celsius.

Line 104 and 114. Check the font of Celsius with spaces.

Line 110. Spaces should be: 4 Celsius to.

Line 156. What is quality of seedling? This word appeared many times both in the title and the text, so please explain in the introduction what character in quality of young seedling is needed to make a contribution to the improvement of seedling production?

Line 162, please define “the specific leaf”.

Line 230. Please cite a reference Nagel et al. 2006 with the number 14 or another.

In discussion. Please add more content to discussion and account for the impact of the report to seedling production. Please cite recent examples regarding tomato plants.

Author Response

(All the changes were highlighted in yellow in the manuscript)

Authors investigated in this article the influence of the level of light intensity of supplemental LEDs lighting on quality of grafted tomato seedlings. They evaluated shoot growth and expression of photosynthetic genes and proteins appeared in grafted tomato seedlings after three levels of supplemental lighting as a treatment for 10 days. Authors concluded that the quality of grafted tomato seedlings was enhanced by the treatment with the optimal PPFD. The findings look helpful for grower of tomato; however, more information should be described in materials and methods to clarify their availability and reproducibility of the result. The meaning of a word, quality, of grafted tomato seedlings was obscure. What is required for grafted tomato seedling to develop well or mature? I could not understand if the assessment of the quality were provided sufficiently with the data appeared in the manuscript. Therefore, the manuscript would be improved if authors could address concerns raised below.

1. In plant materials, how do I confirm that two cultivars of scions purchased and used for grafting to connect with rootstocks had the same stage at plant development?

The characters of the two tomato cultivars were similar, also the sowing time and growing environment were all the same. The related description has been added in line 80 83.

2. In light treatments, the condition of healing and the initial state of grafted seedling before cultivation in a greenhouse with light treatments should be stated.

The condition of healing and the initial state of grafted seedling have been added in line 83-88.

3. In light intensity. How much total PPFD did grafted seedlings gain from natural and supplemental lighting? How far from seedlings were LEDs set in a greenhouse, a distance between seedlings and LEDs, and where were PPFD measured at?

The total PPFD been added as daily light integral (DLI) in line 93-96. The distance between seedlings and LEDs, and where the light intensity measured also have been added in line 96-97.

4. Line 80 and 82, spaces should be inserted between the number and unit: 16 h/day, 25/35 Celsius.

The whole manuscripts were checked for the same mistakes.

5. Line 104 and 114. Check the font of Celsius with spaces.

The whole manuscripts were checked for the same mistakes.

6. Line 110. Spaces should be: 4 Celsius to.

The whole manuscripts were checked for the same mistakes.

7. Line 156. What is quality of seedling? This word appeared many times both in the title and the text, so please explain in the introduction what character in quality of young seedling is needed to make a contribution to the improvement of seedling production?

The description of seedling quality has been added in line 54-58.

8. Line 162, please define “the specific leaf”.

The define of “the specific leaf” has been added at line 177.

9. Line 230. Please cite a reference Nagel et al. 2006 with the number 14 or another.

It has been corrected as number 14.

10 In discussion. Please add more content to discussion and account for the impact of the report to seedling production. Please cite recent examples regarding tomato plants.

Related content has been added in line 236-246. (Green highlight).

Round  2

Reviewer 1 Report

The authors cite many old references (1989, 1990, 1992, 1993, 1994, 1995 ...) but only a few papers from the past 5 years.

Obsolete information has no citation potential. Thanks to new experimental approaches, we know about regulation mechanisms much more than 20 years ago. We cannot ignore them, but we must accept new, up-to-date information.

I recommend potentially using publications dealing with radiation effects in shade and sun plants, organization and function of photosynthetic apparatus, regulatory mechanisms, photoinhibition and photoprotection of PSII and PSI. I invite authors to use some of them:

Changes in morphology, chlorophyll fluorescence performance and Rubisco activity of soybean in response to foliar application of ionic titanium under normal light and shade environment. Science of Total Environment 658, 2019, 626-637; https://doi.org/10.1016/j.scitotenv.2018.12.182

Photosynthetic responses of sun- and shade-grown chlorophyll b deficient mutant of wheat. JOURNAL OF CENTRAL EUROPEAN AGRICULTURE, 2016, Vol: 17, Issue: 4, 950-956, DOI: 10.5513/JCEA01/17.4.1797

Author Response

(All the changes were highlighted in a green color in the manuscript)

The authors cite many old references (1989, 1990, 1992, 1993, 1994, 1995 ...) but only a few papers from the past 5 years.

Obsolete information has no citation potential. Thanks to new experimental approaches, we know about regulation mechanisms much more than 20 years ago. We cannot ignore them, but we must accept new, up-to-date information.

I recommend potentially using publications dealing with radiation effects in shade and sun plants, organization and function of photosynthetic apparatus, regulatory mechanisms, photoinhibition and photoprotection of PSII and PSI. I invite authors to use some of them:

Changes in morphology, chlorophyll fluorescence performance and Rubisco activity of soybean in response to foliar application of ionic titanium under normal light and shade environment. Science of Total Environment 658, 2019, 626-637; https://doi.org/10.1016/j.scitotenv.2018.12.182

Photosynthetic responses of sun- and shade-grown chlorophyll b deficient mutant of wheat. JOURNAL OF CENTRAL EUROPEAN AGRICULTURE, 2016, Vol: 17, Issue: 4, 950-956, DOI: 10.5513/JCEA01/17.4.1797

The relatively new and recommend references on radiation effects in shade and sun plants were added in lines 295-296, 331, and 345-350.

Reviewer 2 Report

Line 78. Please rephrase or delete the following: The ‘SS’ is suitable for retarding culture and forcing culture in greenhouse.

Line 501, Is 'ROS' for reactive oxygen species?

Author Response

(All changes are highlighted in a yellow color in the manuscript)

Line 78. Please rephrase or delete the following: The ‘SS’ is suitable for retarding culture and forcing culture in greenhouse.

This was deleted in lines 76-77.

Line 501, Is 'ROS' for reactive oxygen species?

Full name of ROS (reactive oxygen species) was added in lines 321-322.